# Comparative Transcript Profiling of Resistant and Susceptible Tea Plants in Response to Gray Blight Disease

Rongrong Tan [1,2], Long Jiao [1,2], Danjuan Huang [1,2], Xun Chen [1,2], Hongjuan Wang [1,2] and Yingxin Mao [1,2,*]

1   Key Laboratory of Tea Resources Comprehensive Utilization, Ministry of Agriculture and Rural Affairs, Wuhan 430064, China; tanrongrong1981@hbaas.com (R.T.); jiaolongsry@foxmail.com (L.J.); huangdj@hbaas.com (D.H.); chenxun2021@hbaas.com (X.C.); wanghongjuan@hbaas.com (H.W.)
2   Fruit and Tea Research Institute, Hubei Academy of Agricultural Science, Wuhan 430064, China
*   Correspondence: maoyx1980@hbaas.com

**Abstract:** Gray blight disease stands as one of the most destructive ailments affecting tea plants, causing significant damage and productivity losses. However, the dynamic roles of defense genes during the infection of gray blight disease remain largely unclear, particularly concerning their distinct responses in resistant and susceptible cultivars. In the pursuit of understanding the molecular interactions associated with gray blight disease in tea plants, a transcriptome analysis unveiled that 10,524, 17,863, and 15,178 genes exhibited differential expression in the resistant tea cultivar (Yingshuang), while 14,891, 14,733, and 12,184 genes showed differential expression in the susceptible tea cultivar (Longjing 43) at 8, 24, and 72 h post-inoculation (hpi), respectively. Gene Ontology (GO) and Kyoto Encyclopedia of Genes and Genomes (KEGG) enrichment analyses highlighted that the most up-regulated genes were mainly involved in secondary metabolism, photosynthesis, oxidative phosphorylation, and ribosome pathways. Furthermore, plant hormone signal transduction and flavonoid biosynthesis were specifically expressed in resistant and susceptible tea cultivars, respectively. These findings provide a more comprehensive understanding of the molecular mechanisms underlying tea plant immunity against gray blight disease.

**Keywords:** tea plant; gray blight disease; *Pestealotia theae* Steyaert; transcriptome analysis





## 1. Introduction

The tea plant (*Camellia sinensis* (L.) Kuntze, Theaceae) holds a pivotal role as a key economic crop, and tea stands out as one of the most globally consumed beverages, celebrated not only for its delightful flavors but also for its potential health benefits [1–4]. The tender shoots, composed of two terminal leaves and apical buds, are widely recognized for their use in manufacturing various tea types, such as green tea, black tea, and oolong tea [5]. China, leading the world in tea production, demonstrated a consistent upward trajectory from 2015 to 2022, reaching an impressive 3.35 million tons in 2022, marking a 5.88% year-on-year growth and reinforcing China's enduring dominance in the global tea industry. Despite this success, tea plants are confronted with significant biotic and abiotic stresses, affecting various plant parts and resulting in substantial crop losses worldwide, with reported reductions of up to 55% [6–11]. Addressing these challenges through comprehensive research and intervention is imperative for the sustained growth of global tea production.

Among the challenges faced by tea plants, tea gray blight emerges as one of the most serious foliar diseases, caused by *Pestalotiopsis*-like species, predominantly impacting mature and old foliage while also posing a threat to young shoots [12,13]. The initial documentation of tea gray blight occurred in Kagoshima, Japan, in 1973, and it was identified by the name "zonate leaf spot" at the time [14]. It is particularly notable in regions characterized by insufficient drainage, high temperatures, and heightened humidity—typically prevalent during the summer and autumn seasons [12,15]. In tea plantations severely affected by tea gray blight, yield losses can range from 10% to 20% [16,17]. The existing strategies

for preventing and controlling tea gray blight predominantly rely on chemical measures, which unfortunately contribute to environmental pollution, drug resistance, and concerns about food safety. Specifically, chemical fungicides like methyl benzimidazole carbamates (MBCs) and demethylation inhibitors (DMIs), including carbendazim and thiophanate methyl, are commonly employed. In addition to chemical control, alternative approaches involve the use of potential bacteria, such as *Bacillus* spp. and *Pseudomonas* spp., fungi like *Trichoderma* spp., and plant extracts such as tea saponin and golden flower [18–24]. It is imperative to address the limitations and environmental impact of chemical control methods and explore more sustainable options for managing tea gray blight effectively while ensuring environmental and food safety.

Plants, including tea plants, have evolved intricate innate immune systems, finely tuned to combat pathogens through pattern-triggered immunity (PTI) and effector-triggered immunity (ETI)—key defense mechanisms safeguarding plants from a diverse array of diseases [25]. Research investigations have substantiated the inherent resistance of tea plants to *Colletotrichum fructicola*, emphasizing the pivotal involvement of hypersensitive cell death and the initiation of hydrogen peroxide ($H_2O_2$) production facilitated by R genes through mitogen-activated protein kinase (MAPK) cascades [26]. In response to *C. fructicola*, tea plants exhibit the induction of genes associated with catalytic activity, oxidation–reduction processes, cell wall reinforcement, plant hormone signal transduction, and plant–pathogen interaction [27]. The intricate molecular response in tea plants during tea blister blight infection involves the modulation of various genes, concomitant with heightened activation and up-regulation of disease-resistant transcription factors [28]. Furthermore, S genes, derived from either tea leaves or *Pestalotiopsis trachicarpicola*, significantly contribute to defense-related pathways in the tea plant, such as those associated with glycolysis/gluconeogenesis as well as oxidative phosphorylation in the pathogen [29]. The expression of genes related to cell wall biogenesis and the plant hormone signal transduction pathway shows notable responses to *Didymella segeticola*, particularly in enhancing defense mechanisms [30]. The activation of defense-related pathways in tea plants, particularly those associated with plant–pathogen interaction, MAPK signaling, and plant hormone signal transduction, suggests the presence of a crucial mechanism in responding to *Pseudopestalotiopsis vietnamensis* [31]. Additionally, in response to *Pestealotia theae*, tea plants manifest the up-regulation of phenylpropanoid, flavonoid, and lignin biosynthesis, concomitant with the down-regulation of photosynthesis [32]. However, the intricate molecular regulatory mechanisms underlying the response to *P. theae* remain largely unknown in previous tea plant studies, especially concerning their differential responses in resistant and susceptible cultivars, necessitating further investigation for a comprehensive understanding.

In the present study, we employed transcriptome sequencing using trinity on the Illumina HiSeq platform to establish a comprehensive de novo transcriptome assembly database. Our principal aim was to elucidate the molecular underpinnings governing the tea plant's defense mechanisms against gray blight disease across contrasting cultivars exhibiting differential susceptibility, thereby facilitating a comparative analysis to discern variations in regulatory pathways and gene expression profiles. This comprehensive database was generated from tea leaves infected with gray blight disease, sourced from both the resistant cultivar Yingshuang and the susceptible cultivar Longjing 43 at 8, 24, and 72 h post-inoculation (hpi). The resulting dataset encompasses generated and annotated unigenes, offering valuable insights into gene expression profiles, biochemical processes, and regulatory networks associated with the tea plant's immune response against gray blight disease.

## 2. Materials and Methods

### 2.1. Plant Material and Treatment

All three-year-old asexually reproduced tea seedlings (*Camellia sinensis* (L.) Kuntze, Theaceae), including the resistant cultivar Yingshuang and susceptible cultivar Longjing 43, were cultivated in greenhouse facilities by the Tea and Fruit Research Institute of Hubei Academy of Agricultural Sciences. Tea gray blight disease (*Pestealotia theae* Steyaert) was isolated and purified through

tissue and single-spore methods [33]. Initially, the surface was disinfected using alcohol-soaked cotton. Tissue blocks (0.5 cm$^2$) were excised from healthy–infected boundaries, washed thrice with 75% alcohol for 45 s, and rinsed twice with sterile water. Three tissue blocks on a toothpick were placed on potato dextrose agar (PDA) medium. Sealed dishes were incubated at 25 °C with 68% humidity for 7 days. Fungal mycelium tips were purified, cultivated until spore production, and single spore-purified strains were stored on PDA at 4 °C.

For live plant inoculation, the third or fourth leaves of healthy three-year-old potted tea seedlings were surface-disinfected with 75% alcohol. Four to five small holes were made at the designated site, and a 5 mm fungal cake was applied. Inoculated sites were then wrapped with sterile cotton and covered with cling film for moisture retention, and sterile PDA blocks served as controls. The treatments were replicated three times, and the inoculated plants were cultivated in an artificial climate chamber (26–28 °C, humidity: 75–90%).

## 2.2. RNA Isolation and cDNA Library Construction

Tea seedlings, encompassing both resistant and susceptible varieties, underwent inoculation with *P. theae* for durations of 8, 24, and 72 h. Uninfected seedlings were utilized as controls for RNA isolation and subsequent cDNA library construction. Each experimental condition was replicated three times, resulting in a total of 24 libraries. Transcriptome sequencing was carried out using the Illumina NovaSeq 6000 system (Illumina, San Diego, CA, USA) at Novogene Bioinformatics Technology Co. Ltd. in Beijing, China.

For RNA extraction, the TRIzol reagent (Invitrogen, Carlsbad, CA, USA) was employed, following the manufacturer's protocols. Subsequently, total RNA samples underwent DNA removal using the DNA-free™ Kit (Thermo Fisher, Waltham, MA, USA). Following that, cDNA synthesis was performed utilizing the RevertAid First Strand cDNA Synthesis Kit (Thermo Fisher, Waltham, MA, USA).

RNA quality assessment was primarily conducted using the Agilent 2100 Bioanalyzer, which accurately assesses RNA integrity. Following total RNA extraction, ribosomal RNA was removed to isolate mRNA. Next, mRNA fragments underwent random fragmentation using divalent cations in the NEB Fragmentation Buffer, employing either NEB common fragmentation or chain-specific fragmentation protocols. Subsequently, cDNA libraries were constructed. Upon completion of library construction, initial quantification was performed using the Qubit 2.0 Fluorometer, followed by dilution to a concentration of 1.5 ng/µL. The insert size of the libraries was then evaluated using the Agilent 2100 Bioanalyzer (Santa Clara, CA, USA) to ensure that it met expectations. After confirming the desired insert size, qRT-PCR was employed to accurately quantify the effective concentration of the libraries, ensuring that libraries with an effective concentration greater than 2 nM were retained to maintain quality. Once the libraries passed quality control, they were pooled based on their effective concentrations and the desired sequencing data volume for the Illumina platform.

## 2.3. Transcriptomic Data Analysis

RNA quality assessment, cDNA library construction, and Illumina deep sequencing were conducted following established protocols. To eliminate reads containing adapters, poly-N sequences, and low-quality data from the raw reads, the fastp software (version 0.19.7) with default parameters was employed. Quality metrics such as Q20, Q30, GC content, and sequence duplication levels were calculated using high-quality clean data for subsequent analyses. Q20 and Q30 represent the percentage of bases with a Phred quality score of 20 and 30, respectively, which are indicative of sequencing accuracy. GC content provides insights into the nucleotide composition of the sequences, while sequence duplication levels indicate the extent of PCR amplification bias. After data cleaning, the clean reads were aligned to the assembled reference genome (*Camellia sinensis* (L.) O. Ktze) using HISAT2. Gene model annotation files were downloaded directly from the genome website (http://tpdb.shengxin.ren/, accessed on 1 October 2022).

### 2.4. Expression Analysis and Enrichment Analysis

Based on the positional information of gene alignment on the reference genome, the coverage of reads for each gene (including newly predicted genes) across the entire gene region is calculated. Reads with alignment quality values below 10, unpaired alignments, and reads aligning to multiple regions of the genome are filtered out. Subsequently, gene expression quantification is conducted, followed by statistical analysis to identify genes exhibiting significantly different expression levels across distinct conditions. Differential expression analysis of the samples was conducted using the DESeq2 R package (version 1.16.1). Genes meeting the criteria of an adjusted $p$-value < 0.05 and $|log2Fold Changes| \geq 1$, as determined by DESeq2, were designated as differentially expressed, indicating that their expression levels differed by at least twofold. The method used for adjusting $p$-values and controlling the false discovery rate (FDR) is the Benjamini–Hochberg (BH) method. The BH method ranks the test results based on the original $p$-values and then determines a threshold based on the relationship between the ranked $p$-values and the expected FDR. Gene ontology (GO) enrichment analysis of up-regulated differentially expressed genes (DEGs) was implemented by the cluster Profiler R package, using all *C. sinensis* genes as the background in which gene length bias was corrected, and $p$-values were considered significantly enriched among the DEGs. The statistical enrichment of differential expression genes in the Kyoto Encyclopedia of Genes and Genomes (KEGG) pathways was annotated using (http://www.omicshare.com accessed on 1 October 2022) against the KEGG database.

## 3. Results

### 3.1. Transcriptome Sequencing and De Novo Assembly

In order to investigate the transcriptome response to *P. theae* in tea plants, we conducted RNA-Seq analysis, extracting RNA from the resistant (R, Yingshuang) and susceptible (S, Longjing 43) tea cultivars. These cultivars underwent inoculation with *P. theae* at three time points (8, 24, and 72 h). Subsequently, twenty-four cDNA libraries were constructed from leaf samples of the resistant cultivar (R_CK1, R_CK2, R_CK3, R8h_1, R8h_2, R8h_3, R24h_1, R24h_2, R24h_3, R72h_1, R72h_2, R72h_3) and susceptible cultivar (S_CK1, S_CK2, S_CK3, S8h_1, S8h_2, S8h_3, S24h_1, S24h_2, S24h_3, S72h_1, S72h_2, S72h_3). These libraries were then sequenced using an Illumina HiSeq2500 platform. The sequencing and mapping results, summarized in Table 1, offer a comprehensive overview of this study.

To assess the quality of the sequencing data, we systematically computed the GC content and Q20 and Q30 values using the FastQC (version: 0.11.9) software. The analysis revealed a minimum of 37,671,666 reads and raw bases ranging from at least 6.36 G to a maximum of 7.48 G, maintaining a consistent GC content of approximately 43.35% across all sequencing libraries. Additionally, Q20 values were consistently at least 97.59%, and Q30 values consistently exceeded 92.99% in each sample, indicating a consistently high standard of sequencing quality. Following the assembly process, the pristine data underwent precise mapping onto transcripts and unigenes, resulting in an average mapped ratio of 91.38%. These collective findings strongly suggest that the quality of both throughput and sequencing was sufficiently robust to justify further in-depth analysis.

**Table 1.** Summary of the sequence data from RNA sequencing.

| Sample | Raw_Reads | Raw_Bases | Clean_Reads | Clean_Bases | Q20 (%) | Q30 (%) | GC_Content (%) | Total_Map |
|---|---|---|---|---|---|---|---|---|
| R8_1 | 47,836,158 | 7.18 G | 46,990,646 | 7.05 G | 97.92 | 93.77 | 43.99 | 43,106,457 (91.73%) |
| R8_2 | 45,896,190 | 6.88 G | 45,355,332 | 6.80 G | 97.94 | 93.82 | 43.58 | 41,445,397 (91.38%) |
| R8_3 | 44,696,344 | 6.70 G | 44,055,198 | 6.61 G | 97.79 | 93.41 | 43.68 | 40,257,893 (91.38%) |
| R24_1 | 44,590,504 | 6.69 G | 43,848,424 | 6.58 G | 97.78 | 93.39 | 43.76 | 40,173,934 (91.62%) |
| R24_2 | 43,956,812 | 6.59 G | 43,269,504 | 6.49 G | 97.87 | 93.65 | 43.79 | 39,650,336 (91.64%) |
| R24_3 | 44,448,588 | 6.67 G | 43,630,856 | 6.54 G | 97.73 | 93.26 | 43.87 | 40,000,142 (91.68%) |
| R72_1 | 46,196,388 | 6.93 G | 45,286,722 | 6.79 G | 97.88 | 93.65 | 44.26 | 41,524,195 (91.69%) |
| R72_2 | 44,650,290 | 6.70 G | 43,881,554 | 6.58 G | 97.84 | 93.58 | 44.19 | 40,281,436 (91.80%) |
| R72_3 | 48,300,874 | 7.25 G | 47,230,266 | 7.08 G | 97.97 | 93.91 | 44.23 | 43,398,098 (91.89%) |
| RCK_1 | 47,741,662 | 7.16 G | 46,795,572 | 7.02 G | 97.82 | 93.51 | 43.98 | 42,833,170 (91.53%) |
| RCK_2 | 42,398,504 | 6.36 G | 41,196,186 | 6.18 G | 97.78 | 93.43 | 43.98 | 37,671,666 (91.44%) |
| RCK_3 | 49,856,042 | 7.48 G | 48,792,112 | 7.32 G | 97.99 | 93.90 | 43.96 | 44,719,491 (91.65%) |
| S8_1 | 43,279,170 | 6.49 G | 42,579,708 | 6.39 G | 97.62 | 93.05 | 43.93 | 38,929,410 (91.43%) |
| S8_2 | 47,069,498 | 7.06 G | 46,535,888 | 6.98 G | 97.95 | 93.87 | 43.75 | 42,612,101 (91.57%) |
| S8_3 | 45,521,812 | 6.83 G | 45,053,488 | 6.76 G | 97.97 | 93.86 | 43.80 | 41,334,831 (91.75%) |
| S24_1 | 45,228,878 | 6.78 G | 44,574,556 | 6.69 G | 97.66 | 93.16 | 43.90 | 40,876,575 (91.70%) |
| S24_2 | 44,958,270 | 6.74 G | 44,186,372 | 6.63 G | 97.59 | 92.99 | 44.00 | 40,498,489 (91.65%) |
| S24_3 | 45,000,168 | 6.75 G | 44,412,038 | 6.66 G | 97.78 | 93.60 | 43.35 | 40,695,872 (91.63%) |
| S72_1 | 45,973,516 | 6.90 G | 45,265,712 | 6.79 G | 97.59 | 93.03 | 44.20 | 41,596,740 (91.89%) |
| S72_2 | 45,896,566 | 6.88 G | 45,287,070 | 6.79 G | 97.82 | 93.59 | 44.32 | 41,733,775 (92.15%) |
| S72_3 | 46,282,154 | 6.94 G | 45,221,882 | 6.78 G | 97.73 | 93.31 | 44.28 | 41,627,282 (92.05%) |
| SCK_1 | 46,513,380 | 6.98 G | 45,745,486 | 6.86 G | 97.86 | 93.64 | 44.65 | 42,200,136 (92.25%) |
| SCK_2 | 47,398,758 | 7.11 G | 46,513,712 | 6.98 G | 97.93 | 93.78 | 44.82 | 42,966,349 (92.37%) |
| SCK_3 | 47,369,596 | 7.11 G | 46,569,902 | 6.99 G | 97.86 | 93.63 | 44.58 | 42,961,937 (92.25%) |

Note: R8, R24, R72, and RCK indicate that the resistant tea cultivar was inoculated with *Pestealotia theae* at 8, 24, and 72 h post-inoculation (hpi), with no inoculation serving as the control. Similarly, S8, S24, S72, and SCK indicate that the susceptible tea cultivar was inoculated with *P. theae* at 8, 24, and 72 hpi, with no inoculation as the control.

### 3.2. Identification and Analysis of Differentially Expressed Genes

A differential expression analysis was meticulously conducted, employing specific criteria: a $|\log2(\text{Fold Change})| \geq 1$ and a *p*-value (a common form of false discovery rate (FDR)) $\leq 0.05$. The comprehensive evaluation of gene expression levels revealed distinct clustering patterns among treatment groups for the two tea cultivars at different time points after inoculation with *P. theae* in a principal component analysis (PCA) of the biological replicates. Along the PC1 and PC2 coordinates, these clusters emphasized significant variations in gene expression, providing a clear visualization of treatment-specific effects within the context of the two tea cultivars under investigation (Figure 1a). Subsequently, Venn diagram software (http://www.biovenn.nl/, accessed on 6 February 2023) was utilized to pinpoint commonly expressed genes in both the resistant and susceptible tea cultivars. The outcomes revealed the identification of a total of 38,737 and 38,869 common genes in the resistant and susceptible tea cultivars, respectively. Additionally, at 8, 24, and 72 h post-inoculation (hpi), 703, 677, and 575 genes were specifically detected in the resistant tea cultivar, while 692, 687, and 682 genes were identified in the susceptible tea cultivar, respectively. Moreover, a total of 38,737 shared genes were detected in the resistant tea cultivar, while 38,869 genes were identified in the susceptible tea cultivar (Figure 1b,c). This in-depth analysis provides valuable insights into the shared and unique gene expression patterns between the two tea cultivars at different time points, significantly enhancing our understanding of their distinct responses exhibited by the tea cultivars in response to *P. theae* infection, shedding light on potential mechanisms underlying resistance or susceptibility.

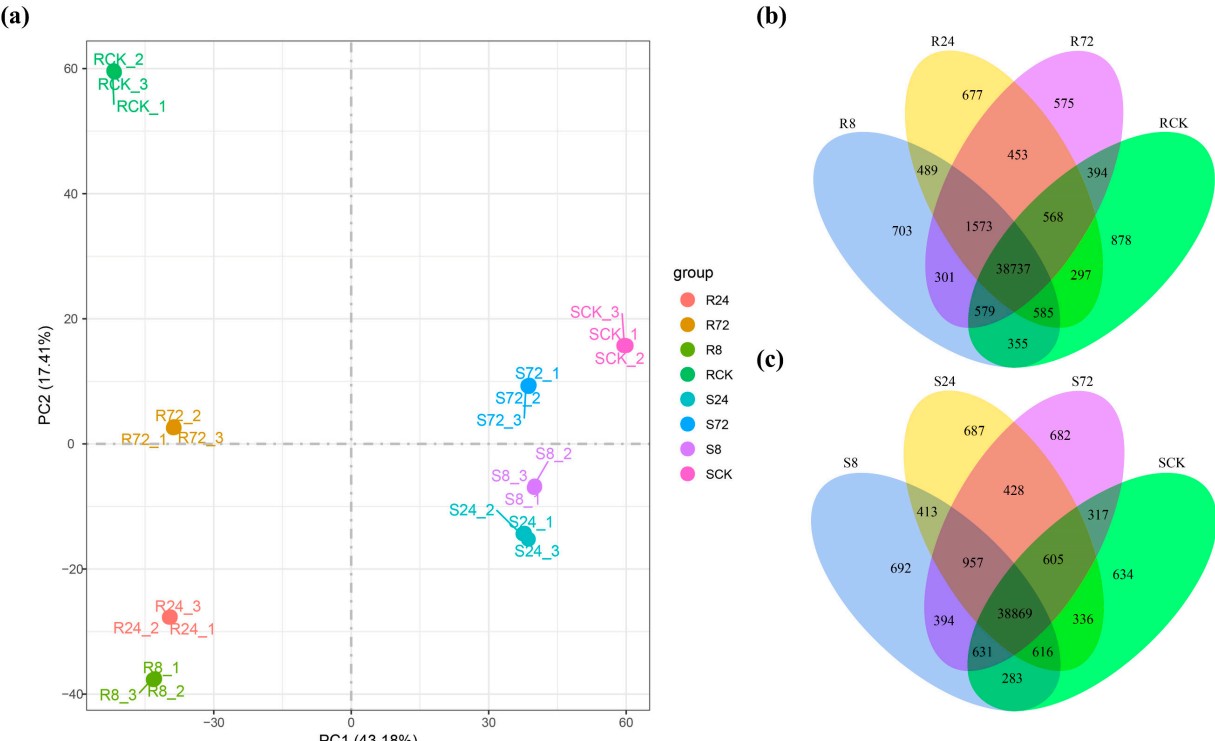

**Figure 1.** The principal component analysis (PCA) and Venn diagram analysis of RNA-seq data. (**a**) PCA analysis highlights discernible differences among treatments, with each point representing one of the three biological replicates corresponding to each treatment (resistant vs. susceptible cultivars) at the tested time points. (**b**,**c**) Venn diagrams illustrate the shared and unique sets of differentially expressed genes (DEGs) identified in the resistant and susceptible tea cultivars. R8, R24, R72, and RCK indicate that the resistant tea cultivar was inoculated with *Pestealotia theae* at 8, 24, and 72 h post-inoculation (hpi), with no inoculation serving as the control. Similarly, S8, S24, S72, and SCK indicate that the susceptible tea cultivar was inoculated with *P. theae* at 8, 24, and 72 hpi, with no inoculation as the control.

### 3.3. DEGs, KEGG, and GO Enrichment Analysis of Resistant Tea Cultivar

To comprehensively investigate gene expression changes during various infection stages of *P. theae* in the resistant tea cultivar, we identified the DEGs by comparing the transcriptomes of R8, R24, R72, and RCK (control without inoculation). In the R8 group, we observed 10,524 DEGs, comprising 4624 up-regulated and 5900 down-regulated genes. For the R24 group, 17,863 DEGs were identified, including 7985 up-regulated and 9878 down-regulated genes (Figure 2a,b and Tables S1 and S2). Additionally, the R72 group revealed a total of 15,178 genes, with 6205 upregulations and 8973 downregulations (Figure 2c and Table S3). Notably, it is observed that the number of up-regulated genes initially increases and then decreases over time, suggesting dynamic changes in the host response to *P. theae* infection in the resistant tea cultivar. The initial rise in the count of differentially expressed genes reflects the activation of immune resistance mechanisms in response to early-stage infection by tea gray blight. Conversely, the subsequent decline in the number of differentially expressed genes indicates the establishment of a stable resistance state. These findings provide valuable insights into the temporal dynamics of gene expression during the infection stages, contributing to a deeper understanding of the molecular mechanisms underlying the tea plant's response to *P. theae*.

To elucidate the potential mechanisms underlying the response of resistant tea cultivars to *P. theae*, KEGG and GO enrichment analyses of up-regulated genes were conducted. A visual representation of the top 20 pathways was portrayed through a bubble plot. At 8 hpi, the KEGG analysis revealed pathways related to photosynthesis, circadian rhythm, and secondary metabolism (nitrogen metabolism, glyoxylate and dicarboxylate metabolism, amino acid metabolism) (Figure 2d). Moving to 24 hpi, the enrichment extended to pathways associated with photosynthesis, DNA replication, the spliceosome, and oxidative phosphorylation (Figure 2e). In contrast, at 72 hpi, there was a shift towards pathways such as ribosomes, the spliceosome, RNA degradation, and RNA transport, indicating dynamic changes in biological processes over time (Figure 2f). Additionally, the GO enrichment analysis highlighted molecular functions enriched in oxidoreductase activity and ion transmembrane transporter activity. At 8 hpi, biological processes such as photosynthesis and DNA-templated transcription were enriched, along with cellular components like thylakoids and the photosynthetic membrane (Figure 2g). By 24 hpi, molecular functions shifted towards RNA binding and catalytic activity, with biological processes encompassing amide biosynthetic processes, cellular responses to stimuli, and cellular components including thylakoids and the photosynthetic membrane (Figure 2h). At 72 hpi, RNA binding and structural molecule activity were prominent in biological processes, along with significant enrichment in peptide biosynthetic and metabolic processes, translation, and cellular components such as ribosomes and membrane-enclosed lumen (Figure 2i). In summary, these results reveal temporal dynamics in pathway activation and molecular function, highlighting the intricate adaptation mechanisms underlying plant defense responses. The transition observed from early defense responses to the subsequent regulation of gene expression and cellular metabolism reflects the gradual adaptation and optimization of resistance mechanisms at different time points during the infection process.

### 3.4. DEGs, KEGG, and GO Enrichment Analyses of Susceptible Tea Cultivar

In the S8, S24, and S72 groups, a total of 14,891, 14,733, and 12,184 DEGs were identified, comprising 7331, 7449, and 5662 up-regulated genes and 7560, 7234, and 6552 down-regulated genes, respectively (Figure 3a–c and Tables S4–S6). Interestingly, the count of DEGs during susceptible tea cultivar *P. theae* infestation exhibited marginal fluctuations at each time point but displayed a slight decrease compared to DEGs during the resistant tea cultivar's *P. theae* infestation. Interestingly, there was a significant involvement of photosynthesis, flavonoid biosynthesis, and secondary metabolism pathways, including amino sugar and nucleotide sugar metabolism, starch and sucrose metabolism, and amino acid metabolism in the response of the susceptible tea cultivar to *P. theae* infestation, particularly at 8 hpi (Figure 3d). At 24 hpi, oxidative phosphorylation and secondary

metabolism pathways, encompassing starch and sucrose metabolism and amino sugar and nucleotide sugar metabolism, were notably enriched (Figure 3e). Furthermore, at 72 hpi, the amino sugar and nucleotide sugar metabolism, starch and sucrose metabolism, endocytosis, and photosynthesis pathways displayed enrichment (Figure 3f).

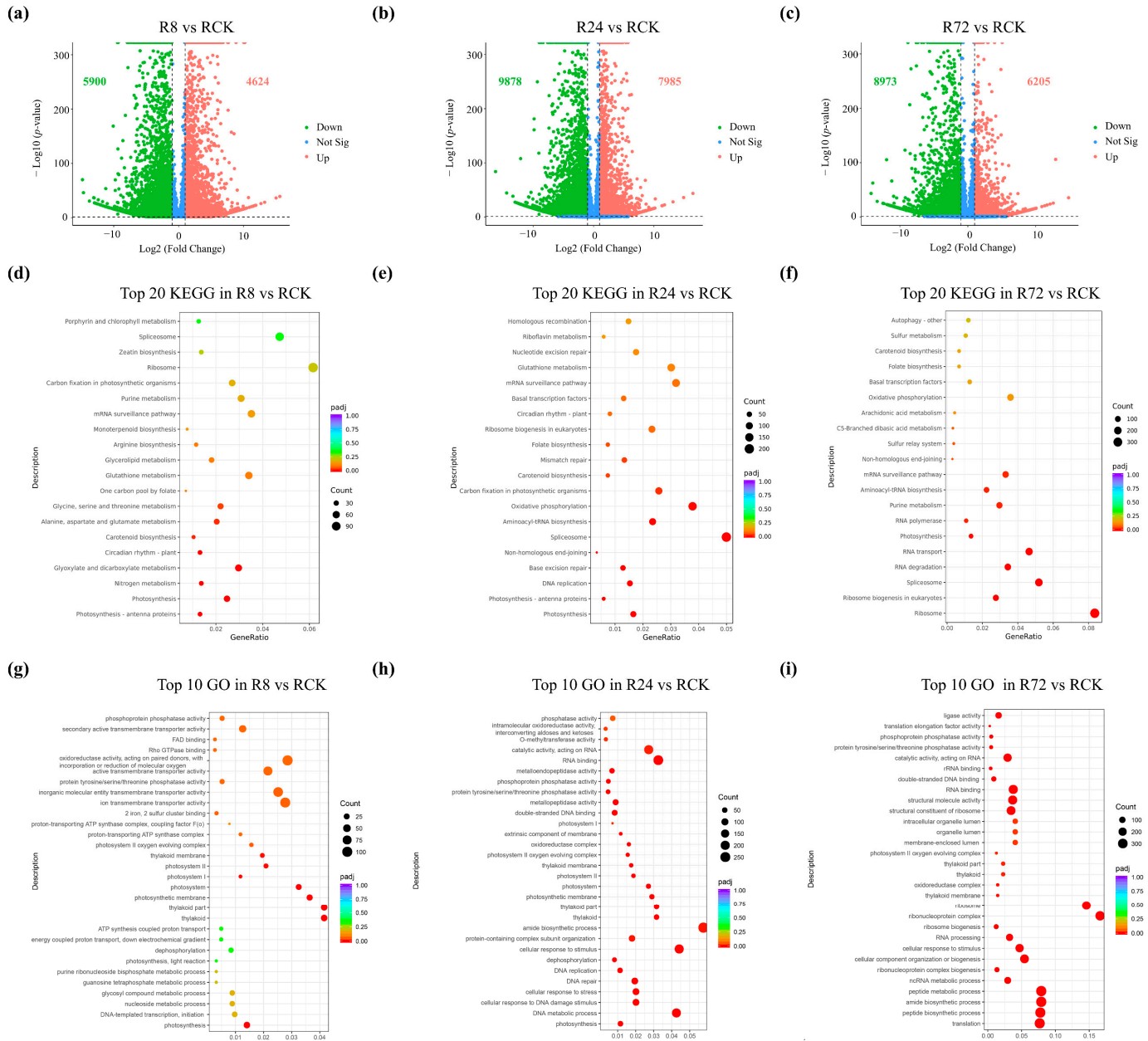

**Figure 2.** Differentially expressed genes (DEGs), Gene Ontology (GO), and Kyoto Encyclopedia of Genes and Genomes (KEGG) enrichment analyses of the resistant tea cultivar inoculated with *P. theae* at 8, 24, and 72 hpi. (**a**–**c**) The volcano plot depicting DEGs of resistant tea cultivar inoculated with *P. theae* at 8, 24, and 72 hpi, respectively. Up-regulated genes are represented by red points, and down-regulated genes are indicated by green points. (**d**–**f**) The top 20 KEGG enrichment pathways identified for up-regulated genes at 8, 24, and 72 hpi, respectively. (**g**–**i**) The top 10 GO enrichment pathways identified for up-regulated genes at 8, 24, and 72 hpi, respectively. The size of each point corresponds to the number of genes, while the color of the points represents the p.adjust (padj).

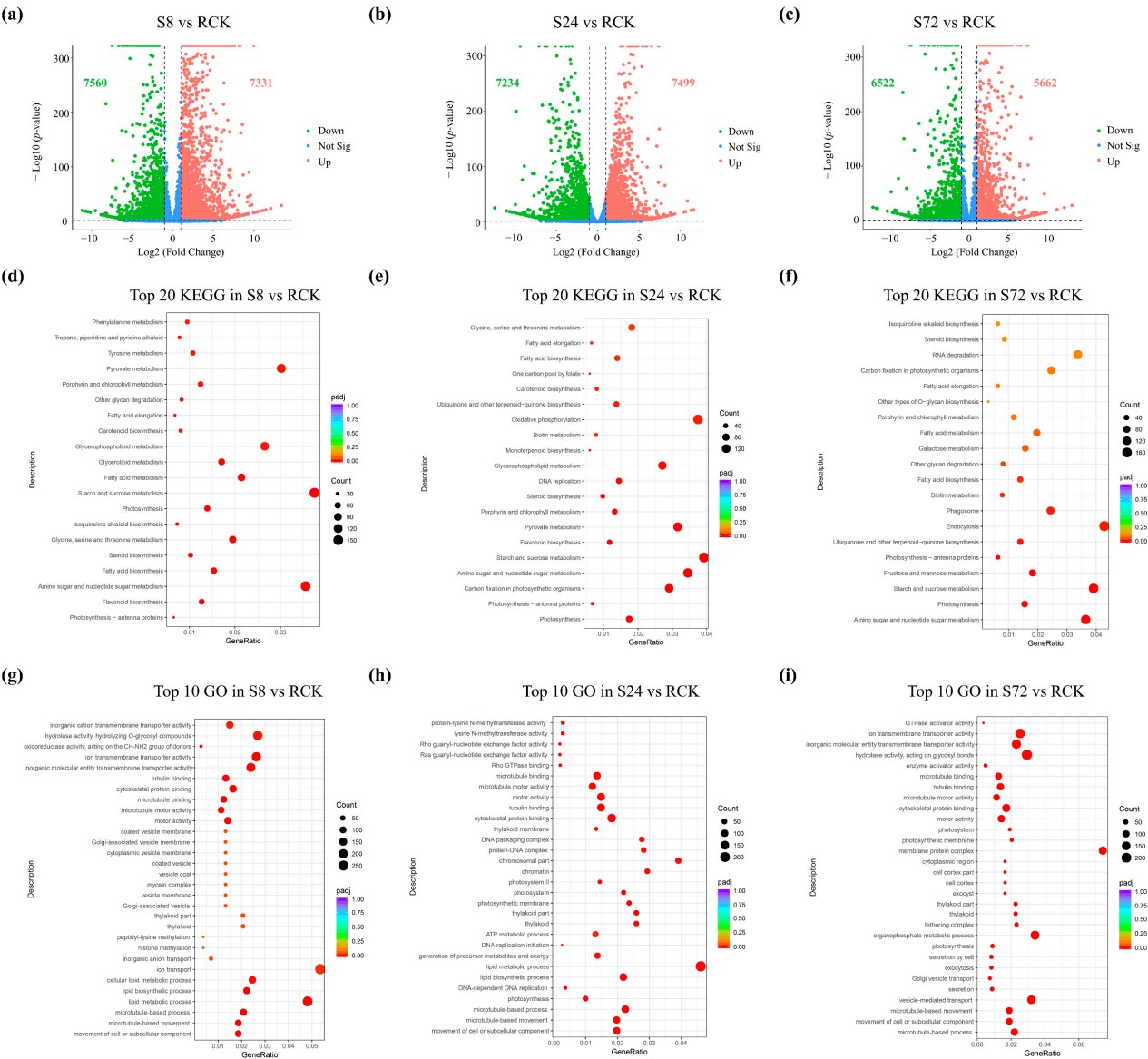

**Figure 3.** DEGs, KEGG, and GO enrichment analyses of the susceptible tea cultivar inoculated with *P. theae* at 8, 24, and 72 hpi. (**a–c**) The volcano plot displays the DEGs of the susceptible tea cultivar inoculated with *P. theae* at 8, 24, and 72 hpi, respectively. Up-regulated genes are marked by red points, while down-regulated genes are represented by green points. (**d–f**) The top 20 KEGG enrichment pathways in analysis of up-regulated genes at 8, 24, and 72 hpi, respectively. (**g–i**) The top 10 GO enrichment pathways identified for up-regulated genes at 8, 24, and 72 hpi, respectively. The size of each point corresponds to the number of genes, while the color of the points represents the p.adjust (padj).

Moreover, the GO enrichment analysis unveiled significant molecular functions, including hydrolase activity and ion transmembrane transporter activity. In the biological processes, there was enrichment in ion transport and lipid metabolic processes, while the cellular components showed enrichment in thylakoid and Golgi-associated vesicles at 8 hpi (Figure 3g). Contrastingly, at 24 hpi, molecular functions shifted towards binding activities such as cytoskeletal protein binding, tubulin binding, and microtubule binding, along with enriched transferase activity. Biological processes encompassed lipid metabolic processes and microtubule-based activities, while cellular components featured thylakoids and the photosystem (Figure 3h). By 72 hpi, biological processes included hydrolase activity and binding (cytoskeletal protein binding, tubulin binding, and microtubule binding), along with photosynthesis and microtubule-based movement (Figure 3i). These results offer a

detailed understanding of the intricate molecular responses in the susceptible tea cultivar during *P. theae* infestation. Furthermore, within the broader field of plant–pathogen interactions, the identification of differential gene expression patterns in tea cultivars during *P. theae* infestation provides valuable insights into the underlying molecular mechanisms of host response. Understanding how specific metabolic pathways, such as photosynthesis, flavonoid biosynthesis, and secondary metabolism, are modulated in response to pathogen attack can contribute to the development of targeted strategies for enhancing plant resistance against pathogens. Additionally, elucidating the dynamic changes in gene expression over time, as observed at 8 hpi, 24 hpi, and 72 hpi, sheds light on the temporal aspects of the plant's defense responses and adaptation strategies. This knowledge can inform future research aimed at engineering tea cultivars with improved resistance to *P. theae* and other pathogens, ultimately contributing to sustainable tea production practices.

*3.5. Comparison of DEGs, KEGG, and GO Enrichment Analyses between Resistant and Susceptible Tea Cultivars*

In the comparative analysis between resistant and susceptible tea cultivars, a total of 19,710 DEGs were identified in the R8 vs. S8 groups, 18,205 DEGs in the R24 vs. S24 groups, and 17,134 DEGs in the R72 vs. S72 groups. These DEGs comprised 9985, 9331, and 8647 up-regulated genes, as well as 9728, 8874, and 8487 down-regulated genes in the R8 vs. S8, R24 vs. S24, and R72 vs. S72 groups, respectively (Figure 4a–c and Tables S7–S9). Despite marginal changes in the count of DEGs at each time point, there was a significant increase compared to DEGs identified separately in the resistant and susceptible tea cultivars during *P. theae* infestation. In terms of the KEGG pathway enrichment analysis, the comparison between resistant and susceptible tea cultivars during *P. theae* infestation revealed the significant involvement of ribosomes, the MAPK signaling pathway, and plant–pathogen interaction at 8 hpi (Figure 4d). At 24 hpi, enrichment was observed in ribosomes, glutathione metabolism, RNA transport, and the citrate cycle (TCA cycle) (Figure 4e). Subsequently, at 72 hpi, glutathione metabolism, ribosomes, the spliceosome, and the RNA transport pathway displayed enrichment (Figure 4f).

For the GO enrichment analysis, up-regulated genes were found to be enriched in pathways involving translation, the peptide biosynthetic process, metabolic processes, ribosomes, binding, and transcription regulator activity at 8 hpi (Figure 4g). At 24 hpi, enrichment was observed in pathways such as amide biosynthetic processes, translation, multicellular organism processes, ribosomes, binding, and the structural constituent of ribosomes (Figure 4h). By 72 hpi, ribonucleoprotein complex biogenesis, RNA processing, ribonucleoprotein complex, ribosomes, binding, and transferase activity were enriched (Figure 4i). These comprehensive analyses provide valuable insights into the differential gene expression, pathway enrichments, and functional categories between resistant and susceptible tea cultivars during *P. theae* infestation at different time points, contributing to a better understanding of the molecular mechanisms underlying resistance and susceptibility responses in tea plants.

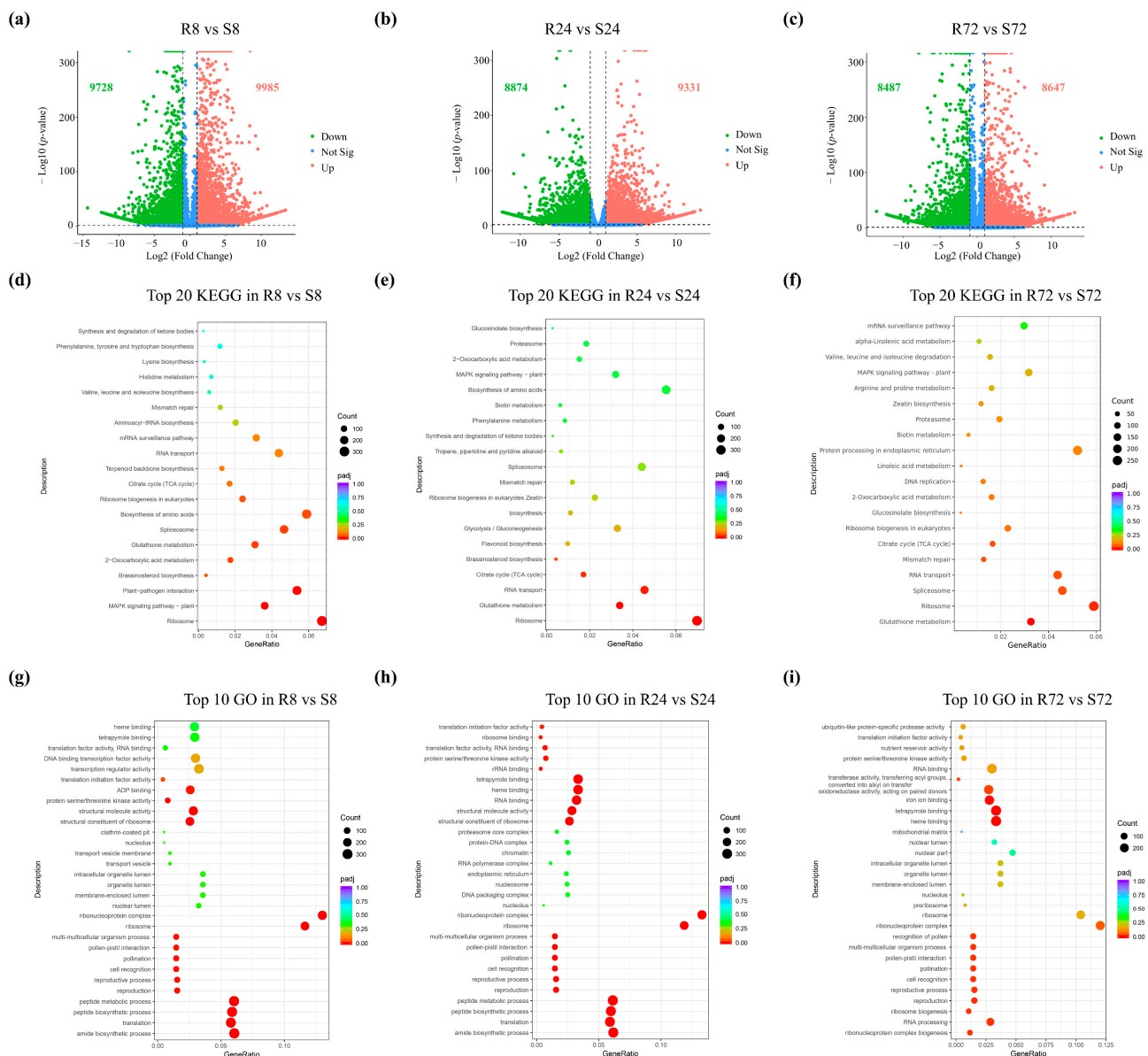

**Figure 4.** DEGs, KEGG, and GO enrichment analyses of the resistant and susceptible tea cultivars inoculated with *P. theae* at 8, 24, and 72 hpi. (**a**–**c**) The volcano plot for DEGs of resistant and susceptible tea cultivars inoculated with *P. theae* at 8, 24, and 72 hpi, respectively. Up-regulated genes are denoted by red points, while down-regulated genes are represented by green points. (**d**–**f**) The top 20 KEGG enrichment pathways for up-regulated genes at 8, 24, and 72 hpi, respectively. (**g**–**i**) The top 10 GO enrichment pathways identified for up-regulated genes at 8, 24, and 72 hpi, respectively. The size of each point corresponds to the number of genes, while the color of the points represents the p.adjust (padj).

## 4. Discussion

Tea, a vital crop esteemed for its economic significance, faces a persistent threat from gray blight disease, posing substantial challenges to global tea crop yields. Despite their economic importance, the precise defense mechanisms employed by tea plants against this disease have remained elusive. Previous studies have shed light on the sophisticated innate immune systems of tea plants, as extensively studied in model plants facing various pathogens such as *Pseudopestalotiopsis* sp., *Colletotrichum* sp., and *Exobasidium vexans* [34]. Understanding the molecular intricacies of plant immunity is crucial for identifying key genes that regulate plant defense. Transcriptome analysis, particularly in non-model species like tea plants lacking sequenced genomes, has emerged as a pivotal tool for functional

gene research. Shi et al. (2011) laid the foundation by employing high-throughput Illumina RNA-seq, establishing a valuable public information platform for gene expression, genomics, and functional genomic studies in *Camellia sinensis* [35].

In our current study, we meticulously conducted a comprehensive transcriptome analysis on 24 libraries derived from the resistant tea cultivar Yingshuang and the susceptible cultivar Longjing 43. These cultivars underwent *P. theae* inoculation for varying durations (8, 24, and 72 h), ensuring the reliability of results through the cultivation of three-year-old asexually reproduced tea seedlings in a controlled greenhouse environment. The analysis yielded a substantial total of 162.34 G clean reads, with a robust total mapping efficiency of at least 91.38% and a Q30 score of at least 92.99%. Furthermore, we utilized a threshold of $p$-values < 0.05 in conjunction with $|\log2\text{Fold Changes}| \geq 1$ as criteria for filtering differentially expressed genes. This approach ensures the identification of genes undergoing significant and reliable changes in expression levels. Specifically, when both criteria are met, indicating a $p$-value < 0.05 and $|\log2\text{Fold Changes}| \geq 1$, it suggests that the expression levels of genes differ by at least twofold between conditions. Such discrepancies are of notable biological relevance as they may signify genuine effects. Moreover, employing this stringent threshold assists in identifying genes with robust expression changes, thereby enhancing their potential functional implications within this study's context. The differential expression analysis revealed a significant number of DEGs in both the resistant and susceptible tea cultivars at different time points following *P. theae* inoculation. Specifically, the resistant tea cultivar exhibited 10,524, 17,863, and 15,178 DEGs, including 4624, 7985, and 6205 up-regulated DEGs at 8, 24, and 72 hpi, respectively. Similarly, the susceptible tea cultivar displayed a total of 14,891, 14,733, and 12,184 DEGs, including 7331, 7499, and 5622 up-regulated DEGs at 8, 24, and 72 hpi, respectively. The fluctuation in the number of up-regulated genes following *P. theae* infection, initially increasing and later decreasing, suggests a dynamic and multifaceted biological response. The initial surge likely signifies a rapid activation of defense mechanisms, while the subsequent decline may indicate the establishment of more stable and effective defense pathways. This dynamic gene expression pattern reflects the intricate and evolving nature of plant responses to pathogen invasion over time.

Our subsequent analyses, encompassing KEGG and GO enrichment, were centered on the up-regulated transcripts observed at each stage of disease progression in both resistant and susceptible tea cultivars following *P. theae* inoculation. In the early stage of *P. theae* infection, both the resistant and susceptible tea cultivars displayed an induction of diverse biological processes. For the KEGG pathway analysis in the resistant tea cultivar, enrichment was observed in pathways such as photosynthesis, nitrogen metabolism, glyoxylate and dicarboxylate metabolism, circadian rhythm—plant, and carotenoid biosynthesis. Conversely, in the susceptible tea cultivar, enriched pathways included photosynthesis, amino sugar and nucleotide sugar metabolism, pyruvate metabolism, and flavonoid biosynthesis. Furthermore, in the GO enrichment analysis, the results revealed enrichment in processes such as photosynthesis and nucleoside metabolic processes and components like thylakoids and the photosynthetic membrane in the resistant tea cultivar. Additionally, oxidoreductase activity and active transmembrane transporter activity were enriched. On the other hand, in the susceptible tea cultivar, enrichment was observed in processes like lipid metabolic processes and microtubule-based processes. Components such as binding and microtubule motor activity were enriched, along with activities like ion transmembrane transporter activity and oxidoreductase activity. Notably, pathways associated with photosynthesis and secondary metabolism stood out in terms of enrichment during the early stage of *P. theae* infection. *PsaD*, *PsaL*, and *PsaK* in photosystem I, along with *PsbO*, *PsbP*, and *PsbQ* in photosystem II, actively participate in photosynthetic pathways. These secondary metabolism pathways included starch and sucrose metabolism, as well as amino sugar and nucleotide sugar metabolism. For example, phosphoglucomutase genes like *PGM2* and *PGM3*, as well as starch synthase genes *SS1*, *SS2*, *SS3*, and *SS4*, demonstrated heightened expression levels. It is noteworthy that these enriched pathways share similarities with the resistance mechanisms documented in previous studies in response to *Colletotrichum* sp. infection in the tea plant [36]. Photosynthesis stands

as a fundamental and universal process that is crucial for the survival of plants, while immune defense plays a pivotal role in adapting to the growth environment. Numerous studies have underscored the intricate interconnection between these two vital processes within a complex network [37–41]. Secondary metabolism pathways emerge as particularly crucial in tea plant resistance to diseases. Research indicates that modulation of the caffeine metabolism pathway reduces fungal tolerance to caffeine and impairs fungal virulence during tea plant infection with *Colletotrichum camelliae* [34,42]. Furthermore, the presence of three primary metabolites in tea—caffeine, theanine, and polyphenols—has been linked to disease resistance properties. Noteworthy studies report that externally applying caffeine can enhance resistance to viruses and bacteria in tobacco plants [43,44]. Interestingly, our KEGG enrichment analysis of up-regulated genes unveiled specific enrichments in plant signal transduction pathways, specifically for salicylic acid, jasmonic acid, and abscisic acid, with a notable emphasis in the resistant tea cultivar (Figure S1). These pathways are well-recognized components of the plant's defense signaling network, signifying a robust and targeted response to *P. theae* infection in the resistant cultivar. This aligns with prior studies that have reported the activation of plant hormone signal transduction, defense-related enzymes, resistance genes, and defense-related transcription factors in the interaction between tea plants and *E. vexans* [45,46]. In contrast, in the susceptible tea cultivar, the flavonoid biosynthesis pathway emerged as a distinctive enrichment. Flavonoids, renowned for their diverse properties encompassing antibacterial, antiviral, antioxidant, anti-inflammatory, antimutagenic, and anticarcinogenic activities, represent a crucial group of secondary metabolites in plant defense [47–51]. The early induction of diverse biological processes in both resistant and susceptible tea cultivars during the early stage of *P. theae* infection lays the foundation for subsequent defense mechanisms, highlighting the interconnectedness between photosynthesis, secondary metabolism pathways, and immune defense responses in tea plants.

Continuing to 24 hpi, the photosynthesis, oxidative phosphorylation, and secondary metabolism pathways continued to play pivotal roles in the infection process with *P. theae*. For the KEGG analysis, pathways such as photosynthesis, oxidative phosphorylation, glutathione metabolism, spliceosome, and DNA replication were enriched in the resistant tea cultivar. Alternatively, in the susceptible tea cultivar, enriched pathways included photosynthesis, amino sugar and nucleotide sugar metabolism, starch and sucrose metabolism, flavonoid biosynthesis, and oxidative phosphorylation. In the GO analysis, processes such as photosynthesis, amide biosynthetic processes, thylakoids, the photosynthetic membrane, RNA binding, and catalytic activity were enriched in the resistant tea cultivar. Conversely, in the susceptible tea cultivar, enrichment was observed in processes like photosynthesis, microtubule-based processes, lipid metabolic processes, thylakoids, the photosynthetic membrane, as well as in binding and microtubule motor activity. These pathways suggest a sustained effort by both resistant and susceptible tea cultivars to address the pathogen's presence. Additionally, at this stage, flavonoid biosynthesis and circadian rhythm-related genes exhibited specific expression patterns, with distinct responses in observed in the susceptible and resistant tea cultivars, respectively. Remarkably, at 72 hpi, there was an activation of pathways related to ribosomes, RNA degradation, and secondary metabolism, including starch and sucrose metabolism and amino sugar and nucleotide sugar metabolism. Specifically, in the KEGG pathways analysis, the resistant tea cultivar showed enrichment in pathways such as ribosomes, the spliceosome, RNA degradation and transport, and purine metabolism, whereas the susceptible tea cultivar exhibited enrichment in pathways including amino sugar and nucleotide sugar metabolism, photosynthesis, starch and sucrose metabolism, endocytosis, and RNA degradation. Regarding the GO results, enrichment in translation, peptide biosynthetic processes, amide biosynthetic processes, peptide metabolic processes, ribosomes, RNA binding, and catalytic activity was observed in the resistant tea cultivar. Conversely, the susceptible tea cultivar showed enrichment in ion transmembrane transporter activity, hydrolase activity, binding, organophosphate metabolic processes, and vesicle-mediated transport. This late-stage activation of ribosome-related pathways suggests an increased demand for protein synthesis, potentially linked to the tea plant's response to the sustained infection. The activation of

ribosome-related pathways at 72 hpi may indicate a crucial role in the plant's defense mechanism, potentially involving the synthesis of specific proteins that contribute to the response against *P. theae*. Additionally, the involvement of RNA degradation pathways may indicate a regulatory mechanism to fine-tune gene expression during the later stages of the infection process. These dynamic changes in pathway activation over the course of infection highlight the complex and multifaceted nature of the tea plant's response to *P. theae*, revealing intricate mechanisms involved in combating the pathogen and adapting to its presence.

In a comparative analysis between resistant and susceptible tea cultivars post *P. theae* inoculation at 8, 24, and 72 hpi, we identified 19,710, 18,205, and 17,134 DEGs, respectively. The KEGG analysis revealed enrichment in pathways such as plant–pathogen interaction, the MAPK signaling pathway (plant), and ribosomes at 8 hpi, while at 24 hpi, pathways including ribosomes, glutathione metabolism, RNA transport, and the TCA cycle were enriched. Similarly, at 72 hpi, the enriched pathways were glutathione metabolism, ribosomes, the spliceosome, and the MAPK signaling pathway (plant). Additionally, the GO analysis showed enrichment in processes like amide biosynthetic processes, translation, peptide metabolic processes, binding, and ribosomes at 8 hpi and 24 hpi. However, at 72 hpi, enrichment was noted in oxidoreductase activity, binding, RNA processing, and ribosome biogenesis. Notably, several pathways, including MAPK signaling, plant–pathogen interaction, ribosomes, glutathione metabolism, RNA transport, and the TCA cycle, exhibited activation in the tea plant's response to the advancing infection. In our study, we observed up-regulation of *MAPK1*, *MAPK3*, and *MAPK4*, as well as *PR1* and *WRKY29*, within the *MAPK* signaling pathways. Moreover, during the hypersensitive response, the expression levels of *RPM1*, *RPS2*, *SGT1*, and *HSP90* were reduced. *RPM1* and *RPS2* are genes associated with plant disease resistance, playing crucial roles in signaling pathways that activate defense responses against invading pathogens. *SGT1*, known as Suppressor of G-Two Allele of *Skp1*, is essential for regulating plant immune responses by interacting with various proteins involved in defense signaling. *HSP90*, heat shock protein 90, acts as a molecular chaperone, aiding in the proper folding and stability of proteins, including those involved in immune responses. Numerous prior studies have underscored the pivotal roles of *MPK* genes in plant disease resistance, overseeing multiple defense responses [52–56]. For instance, Kandoth et al. (2007) revealed that the co-silencing of *MPK1* and *MPK2* compromises resistance to *Manduca sexta* herbivory [57]. These findings collectively highlight the complex and dynamic molecular responses of tea plants to pathogenic challenges, providing valuable insights into the mechanisms underlying their resistance strategies. Moreover, understanding these molecular responses could aid in the development of more effective strategies for disease management and crop improvement in tea cultivation.

Our study aimed to elucidate the immunologic mechanisms employed by tea plants to combat gray blight disease, specifically focusing on the immunologic mechanisms revealed by comparing the differential gene expression during the process of *P. theae* infection through transcriptomic analysis in both resistant and susceptible tea cultivars. Key pathways, such as secondary metabolism and photosynthesis, emerged as crucial during *P. theae* infection in both types of cultivars. Moreover, distinctive expression patterns were observed in plant hormone signal transduction pathways and flavonoid biosynthesis, with the former being specifically expressed in susceptible tea cultivars and the latter in resistant tea cultivars. These findings significantly contribute to our understanding of the intricate interplay between *P. theae* and the tea plant, providing a foundation for future research aimed at enhancing disease resistance and overall crop health. The identified pathways and genes represent potent targets for further investigation, ultimately supporting the development of strategies to mitigate the impact of gray blight disease in tea cultivation.

## 5. Conclusions

Our transcriptome analysis revealed significant differences in gene expression between resistant (Yingshuang) and susceptible (Longjing 43) tea cultivars following gray blight infection. We identified genes associated with secondary metabolism, photosynthesis,

oxidative phosphorylation, and ribosome pathways that were up-regulated in response to the disease. Additionally, distinct expression patterns of genes involved in plant hormone signal transduction and flavonoid biosynthesis were observed in resistant and susceptible cultivars. These findings deepen our understanding of tea plant immunity against gray blight disease and provide insights into potential strategies for disease management and crop improvement in tea cultivation.

**Supplementary Materials:** The following supporting information can be downloaded at https://www.mdpi.com/article/10.3390/agronomy14030565/s1, Figure S1: Kyoto Encyclopedia of Genes and Genomes (KEGG) enrichment analysis of all differentially expressed genes (DEGs) in the resistant tea cultivar inoculated with *Pestealotia theae* at 8 h post-inoculation (hpi). Table S1: Differentially expressed genes identified in the resistant tea cultivar inoculated with *Pestealotia theae* at 8 h post-inoculation. Table S2: Differentially expressed genes identified in the resistant tea cultivar inoculated with *Pestealotia theae* at 24 h post-inoculation. Table S3: Differentially expressed genes identified in the resistant tea cultivar inoculated with *Pestealotia theae* at 72 h post-inoculation. Table S4: Differentially expressed genes identified in the susceptible tea cultivar inoculated with *Pestealotia theae* at 8 h post-inoculation. Table S5: Differentially expressed genes identified in the susceptible tea cultivar inoculated with *Pestealotia theae* at 24 h post-inoculation. Table S6: Differentially expressed genes identified in the susceptible tea cultivar inoculated with *Pestealotia theae* at 72 h post-inoculation. Table S7: Comparison of differentially expressed genes in the resistant and susceptible tea cultivars inoculated with *Pestealotia theae* at 8 h post-inoculation. Table S8: Comparison of differentially expressed genes in the resistant and susceptible tea cultivars inoculated with *Pestealotia theae* at 24 h post-inoculation. Table S9: Comparison of differentially expressed genes in the resistant and susceptible tea cultivars inoculated with *Pestealotia theae* at 72 h post-inoculation.

**Author Contributions:** Conceptualization, R.T.; methodology, X.C. and H.W.; software, X.C. and H.W.; validation, D.H.; formal analysis, R.T.; investigation, X.C. and H.W.; resources, H.W.; data curation, L.J. and D.H.; writing—original draft preparation, R.T.; writing—review and editing, L.J. and Y.M.; visualization, D.H.; supervision, L.J. and Y.M.; project administration, Y.M.; funding acquisition, Y.M. All authors have read and agreed to the published version of the manuscript.

**Funding:** This work was supported by the National Key Research and Development Program of China (2021YFD1601100), the China Agriculture Research System of MOF and MARA (CARS-19), and the Innovation Center Fund for Agricultural Science and Technology in Hubei Province of China (2023-620-005-001).

**Data Availability Statement:** The original contributions presented in the study are included in the article/Supplementary Materials, further inquiries can be directed to the corresponding author.

**Conflicts of Interest:** The authors declare no conflicts of interest.

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
