# Peer review of "Comparative Transcript Profiling of Resistant and Susceptible Tea Plants in Response to Gray Blight Disease"

_agronomy, doi:10.3390/agronomy14030565_

Round 1

Reviewer 1 Report

Comments and Suggestions for Authors

The manuscript presents a thorough investigation into the transcriptomic response of tea plants to infection by Pestealotia theae (P. theae), comparing resistant and susceptible cultivars over three time points post-inoculation. The authors utilized RNA-Seq analysis to generate comprehensive data on gene expression profiles, employing stringent criteria for differential expression analysis. The results demonstrate distinct clustering patterns in gene expression between the two cultivars, with significant numbers of differentially expressed genes (DEGs) identified at each time point. Furthermore, pathway enrichment analyses shed light on the molecular mechanisms underlying the response to P. theae infection, highlighting the involvement of pathways related to photosynthesis, secondary metabolism, and plant hormone signaling.

However, despite the thoroughness of the analysis, there are several notable issues with the manuscript. Firstly, while the methodology and results are well-described, the manuscript lacks a clear overarching research question or hypothesis. Without a clear statement of the problem being addressed, the significance of the findings may be somewhat diminished. Additionally, the discussion section could benefit from more detailed interpretation and contextualization of the results within the broader field of plant-pathogen interactions. Moreover, while the manuscript provides valuable insights into the transcriptomic response of tea plants to P. theae infection, it could be strengthened by including validation experiments or functional assays to corroborate the findings and provide further evidence for the identified pathways and mechanisms. Overall, while the manuscript presents valuable data and analysis, addressing these issues would enhance its clarity and impact.

Reviewer 2 Report

Comments and Suggestions for Authors

The manuscript titled "Comparative Transcript Profiling of Resistant and Susceptible Tea Plants in Response to Gray Blight Disease" provides a comprehensive overview of the challenges posed by tea gray blight on the tea plant, focusing on the economic significance of the tea industry, the impact of the disease, and the limitations of current control measures. The introduction effectively sets the stage for the study, emphasizing the need for sustainable and environmentally friendly strategies to address the issue.

The methods section details the experimental design, including the cultivation of tea seedlings, the isolation and purification of the gray blight pathogen, and the procedures for live plant inoculation, RNA isolation, and cDNA library construction. The use of both resistant and susceptible cultivars, along with multiple time points for transcriptome analysis, strengthens the experimental design and enhances the reliability of the findings.

However, some additional information could be included to enhance the clarity and reproducibility of the study. For instance, providing specific details on the statistical methods employed for the differential expression analysis, such as the method used for adjusting P-values and controlling false discovery rates, would be beneficial. Additionally, the detailed protocols for RNA quality assessment, library construction, and sequencing could be provided in supplementary materials for the benefit of researchers attempting to replicate the study.

Furthermore, the manuscript could benefit from a more explicit description of the criteria used to determine differentially expressed genes (DEGs) and their biological significance. The authors should consider providing more context on why the chosen |log2Fold Changes| ≥ 1 threshold is relevant to the study and how it relates to the biological significance of the observed gene expression changes.

Overall, the manuscript is well-written, the results clearly presented and the conclusions are supported by the results.

Reviewer 3 Report

Comments and Suggestions for Authors

MS review:

Comparative transcript profiling of resistant and susceptible tea plants in response to gray blight disease

Rongrong Tan, Long Jiao, Danjuan Huang, Xun Chen, Hongjuan Wang, Yingxin Maо

The peer-reviewed study aims to elucidate the immunological mechanism by comparing differential gene expression during P. theae infection through transcriptomic analysis in both tea-resistant and susceptible cultivars. As is known, gray mold is one of the most destructive diseases affecting tea plants, causing significant damage and loss of productivity. However, the dynamic role of defense genes during gray mold infection remains largely unclear. The research results indicate that secondary metabolism and photosynthesis were decisive for P. theae infection in both types of varieties. Moreover, distinctive expression patterns were observed in plant hormone signaling pathways and flavonoid biosynthesis, with the former specifically expressed in tea-susceptible cultivars and the latter in tea-tolerant cultivars.

Results obtained during the implementation of this work make significant contributions to the understanding of the complex interactions between P. theae and the tea plant, providing a basis for future research aimed at improving disease resistance and overall crop health. The identified pathways and genes represent potential targets for further study, ultimately facilitating the development of strategies to mitigate the effects of botrytis in tea cultivation. These results provide a more complete understanding of the molecular mechanisms underlying tea plant immunity against gray mold.

The work is interesting and relevant. It was carried out using the traditional method, which will allow us to compare this result with other similar studies. The results obtained are important in agricultural research.

Round 2

Reviewer 1 Report

Comments and Suggestions for Authors

The authors fulfill the suggestions.